# CAN STABILITY BE DETRIMENTAL? BETTER GENERALIZATION THROUGH GRADIENT DESCENT INSTABILITIES

## ABSTRACT

Traditional analyses of gradient descent optimization show that, when the largest eigenvalue of the loss Hessian - often referred to as the sharpness - is below a critical learning-rate threshold, then training is 'stable' and training loss decreases monotonically. Recent studies, however, have suggested that the majority of modern deep neural networks achieve good performance despite operating outside this stable regime. In this work, we demonstrate that such instabilities, induced by large learning rates, move model parameters toward flatter regions of the loss landscape. Our crucial insight lies in noting that, during these instabilities, the orientation of the Hessian eigenvectors rotate. This, we conjecture, allows the model to explore regions of the loss landscape that display more desirable geometrical properties for generalization, such as flatness. These rotations are a consequence of network depth, and we prove that for any network with depth $> 1$, unstable growth in parameters cause rotations in the principal components of the Hessian, which promote exploration of the parameter space away from unstable directions. Our empirical studies reveal an implicit regularization effect in gradient descent with large learning rates operating beyond the stability threshold. We find these lead to excellent generalization performance on modern benchmark datasets.

## 1 INTRODUCTION

Deep neural networks are widely successful across a number of tasks, but their generalization performance is dependent on careful choices of hyperparameters which govern the learning process. Gradient descent (including stochastic gradient descent and ADAM (Kingma & Ba, 2017)) is arguably the most widely-used learning algorithm due to its simplicity and versatility. For such methods, the *descent lemma* upper-bounds the choice of learning rate by the local curvature (or sharpness) to guarantee stable optimization trajectories and provable decreases for convex training losses.

Recently, the 'unstable' learning-rate regime has been a focal point for research. Cohen et al. (2022) have demonstrated that in practice, learning rates can go above the stability threshold as determined by the *descent lemma*. Surprisingly, this appears not to destabilize training trajectories as expected, but instead model sharpness continues near the stability limit while training loss improves. Large learning rates have also been known to improve generalization performance. Our work builds on these findings, showing that training instabilities, caused by large learning rates, drive model parameters toward flatter regions of the loss landscape to improve generalization.

The main contribution of our work, detailed in Section 3, proposes that these instabilities are resolved through rotations in the eigenvectors of the loss Hessian. We demonstrate that, for deep neural networks, unstable training causes these eigenvectors to rotate away from their previous orientations, whereas in the stable regime, the orientations are reinforced. These rotations allow the model to explore regions of parameter space with an in-built bias for flatness. We term the accumulation of this implicit regularization effect *progressive flattening*, which we validate empirically.

Throughout our experiments, learning rates allow direct control over the magnitude of these regularization effects. Our empirical study in Section 4 explores the relationship between learning rates and generalization, revealing a clear phase transition where generalization benefits only emerge for learning rates beyond the stability threshold, solidifying the role of instabilities in regularization.

Additionally, we show that starting with large learning rates have long-term benefits in generalization performance, even when learning rates are reduced later in training. However, our findings also challenge the reliability of sharpness as a metric for generalization, as the degree of eigenvector rotation can, in some cases, be a more effective predictor. The code for reproducing the experiments and results in this paper will be made available on GitHub upon publication.

## 2 BACKGROUND

### 2.1 GRADIENT DESCENT AND THE DESCENT LEMMA

We denote the loss function by $L(\theta)$ with parameters $\theta$ that are updated with gradient descent using learning rate $\eta$, i.e. $\theta_{t+1} = \theta_t - \eta\nabla L(\theta_t)$. The so-called *sharpness* $S$ of the loss landscape is typically estimated with the maximum curvature of the loss Hessian, i.e. $S(\theta) = \lambda_{\max}(H(\theta)) = \lambda_{\max}(\nabla^2 L(\theta))$. The *descent lemma* can be stated as:

**Lemma.** *For a convex, $l$-smooth function $L(\theta)$,* $L(\theta_{t+1}) \leq L(\theta_t) - \eta(1 - \frac{\eta l}{2})\|\nabla L(\theta_t)\|_2^2$

The proof uses the co-coercive property of $l$-smooth functions using the results of Baillon & Haddad (1977). The decrease in loss is scaled by the quadratic $\eta(1 - \frac{\eta l}{2})$, which leads to the optimal learning rate $\eta = 1/l$. However, any choice of $0 < \eta < 2/l$ guarantees a decrease in the loss function, allowing convergence to the minima in the *stable* regime of training, leading to a popular rule for $\eta$ selection. On the other hand, choosing $\eta > 2/l$ results in so-called *instabilities*, causing $L$ and $\theta$ to grow without bound. Additionally, when $\eta > 1/l$, parameters *oscillate* about the minima.

Although these bounds are derived from $l$, in practice $\eta$ is chosen without knowing $l$, leading to an empirical stability threshold of $S(\theta)$ given by $S(\theta) \leq 2/\eta$, beyond which training is thought to destabilize. However, Cohen et al. (2022) showed that in practice, training is not destabilized as expected. They identified two important phenomena for gradient descent:

**Progressive Sharpening**: So long as training is stable ($S(\theta) \leq 2/\eta$), gradient descent has an overwhelming tendency to continually increase sharpness.

**Edge of Stability**: Once sharpness reaches the stability limit, it sits at, or just above, the stability threshold. Additionally, although the descent lemma does not guarantee a decrease of training loss, the training loss nonetheless continues to decrease, albeit non-monotonically.

In practice, loss functions extend beyond the quadratic and instabilities manifest as spikes in $\theta$, $L(\theta)$, and $S(\theta)$. Damian et al. (2023) introduced the progressive sharpening factor $\alpha = -\nabla L(\theta) \cdot \nabla S(\theta)$., where a positive $\alpha$ indicates that gradient updates promote sharpening. The authors hypothesized that during instabilities, $\alpha > 0$, meaning progressive sharpening is active and prolongs the instabilities. However, our initial empirical results (see Appendix D) suggest that this effect, in its currently formulation, plays a limited role during instability, and we posit that the primary driver of instability is unstable growth in parameters, driven by $S(\theta) > 2/\eta$ true to the local quadratic approximation.

### 2.2 SHARPNESS AND GENERALIZATION

The notion that solutions with low sharpness, or flat minima, promotes generalization performance is widely accepted. Hochreiter & Schmidhuber (1997) argued, using minimum description length, that flatter solutions are less complex, thereby improving generalization through appeal to *Occam's Razor*. More recently, Keskar et al. (2017) and Jastrzębski et al. (2018) observed that deep neural networks trained with small learning rates tend to generalize poorly because the minima they converged to were narrow. The width of minima is measured through sharpness, $S(\theta)$, often defined as the largest eigenvalue of the loss Hessian. However, Dinh et al. (2017) demonstrated that some output-preserving transformations can lead to arbitrary values of $S(\theta)$, revealing a lack of scale-invariance. While this observation weakens the absolute causality from sharpness to generalization, Jiang et al. (2019) found that sharpness may still serve as a useful indicator of generalization performance.

Modern methods for efficiently computing eigenvalue-vector pairs of the Hessian utilize Pearlmutter (1994)'s trick, which allows evaluation of the Hessian vector product without explicitly forming the Hessian. In our work, we present an implementation in *Jax* (Bradbury et al., 2018) that also employs matrix-power-kernel (MPK) re-orthogonalization (Yamazaki et al., 2024) to enhance numerical

stability. Additionally, our study focuses on the similarity between eigenvectors, which can naturally be compared with cosine-similarity in one-dimension. For comparisons between subspaces formed by multiple eigenvectors, we use the cosine-Grassmanian distance (Ye & Lim, 2016). The code to reproduce the results in this paper will be made available on GitHub upon publication.

## 3 REGULARIZATION THROUGH INSTABILITIES

We study the dynamics of gradient descent during instabilities - a phase of learning driven by unstable growth in parameters and characterized by spikes in $\theta$, $L(\theta)$ and $S(\theta)$. In this section, we introduce a toy problem to demonstrate that unstable parameter growth cause rotations in the eigenvectors of the loss Hessian. Adopting a reductionist approach, we attempt to explain the complex behavior of the overall loss function by analyzing individual terms. To model the dynamics of one such term, we utilize a Diagonal Linear Network (DLN), a simple neural network with non-trivial dynamics, as studied in Pesme et al. (2021). The DLN's multiplicative structure reflects a key feature of depth, which weights are multiplied across vertically-stacked neural layers whenever depth $d > 1$. We focus on a detailed study of a two-parameter DLN, and the derivation for a general $n$-parameter model is provided in Appendix B. We then identify phases of $\eta$ that match up exceedingly well with established stability bounds, using these insights to conjecture how gradient descent instabilities can be resolved through eigenvector rotations. Finally, we show that the cumulative effects of these dynamics over extended periods of training can result in *progressive flattening* of the loss landscape.

### 3.1 ROTATIONS FROM PARAMETER GROWTH - A DLN MODEL

Consider a loss function $L(\Theta)$, where the parameter $\Theta$ is a multiplication of weights $\Theta = \prod_i^n \theta_i$, which is equivalent to a DLN. For any neural network with depth $d > 1$, the network outputs involve terms that are the product of weights across layers. We focus on one such term, $\Theta$, demonstrating that parameter growth induces rotations in the sharpest Hessian eigenvectors. By examining the two-parameter case, where exactly derivations are possible, we obtain key insights into these rotations. The extension to a general $n$-parameter DLN is explored in detail in Appendix B.

Let $L(\Theta)$ be described by $z(\Theta)$, a non-negative convex polynomial with a unique minimum at $\Theta = \theta_1 \theta_2 = 0$, which limits $z(\Theta)$ to even degree polynomials. Additionally, assume that the parameters are not at the minimum, i.e. $\Theta \neq 0$. Writing $z' := \frac{\partial z}{\partial \Theta}$, we get the loss Hessian $H$:

$$
H(\Theta) = \begin{bmatrix} \frac{\partial^2 L}{\partial \theta_1^2} & \frac{\partial^2 L}{\partial \theta_1 \theta_2} \\ \frac{\partial^2 L}{\partial \theta_1 \theta_2} & \frac{\partial^2 L}{\partial \theta_2^2} \end{bmatrix} = \begin{bmatrix} z'' \theta_2^2 & z'' \theta_1 \theta_2 + z' \\ z'' \theta_2 \theta_1 + z' & z'' \theta_1^2 \end{bmatrix}
$$

Additionally, denote the eigenvector-value pairs $(\lambda_i, \boldsymbol{v}_i), i \in \{1, 2\}$ in the basis formed by $\theta$s: $\boldsymbol{v}_i := [w_{i,1}, w_{i,2}]^T$. We solve exactly for $n = 2$ to get $R$, a ratio of coordinates (see Appendix A.1):

$$
R = \left| \frac{w_{1,1}}{w_{1,2}} \right| = \beta + \sqrt{\beta^2 + 1}, \text{ where } \beta = \left| \frac{z''(\theta_2^2 - \theta_1^2)}{2(z' + z''\Theta)} \right| \tag{1}
$$

We note that $R$ is a monotonic function of $\beta$. Practically, $R$ reflects a degree of alignment between the sharpest eigenvector and the sharpest parameter, which we will use to characterize the orientation of the sharpest eigenvector following gradient updates.

Consider an optimization trajectory initialised at $[\theta_1, \theta_2]^T$ using a fixed learning rate $\eta$. Without loss of generality, assume that $\theta_2^2 > \theta_1^2$, which implies that $L(\Theta)$ is more sensitive to to changes in $\theta_1$ than $\theta_2$, or that $\theta_1$ is the sharper parameter. Importantly, we note that $R$ is sign-independent (see Appendix A.1) and is invariant to exchanges in the magnitude of $theta$s. The gradient updates of $\theta$s are:

$$
\Delta \theta_1 = \eta \frac{dz}{d\theta_1} = \eta z' \theta_2; \text{ similarly } \Delta \theta_2 = \eta z' \theta_1
$$

Despite originating from the same loss function $z$, the updates to these parameters vary in scale owing to the multiplicative nature of $\Theta$. This leads to $\Delta \theta_1 > \Delta \theta_2$, leading to instabilities when $\eta$ is too large. Given these updates, the ratio of changes in $\beta$, $\gamma_\beta$ is:

$$
\gamma_\beta := \frac{\beta + \Delta \beta}{\beta} \approx \frac{\gamma_{|\theta_2^2 - \theta_1^2|}}{\gamma_\Theta} = \left| \frac{1 - \eta^2 z'^2}{1 - \eta z' k + \eta^2 z'^2} \right| \text{ where } k = \left( \frac{\theta_1}{\theta_2} + \frac{\theta_2}{\theta_1} \right) > 2 \tag{2}
$$

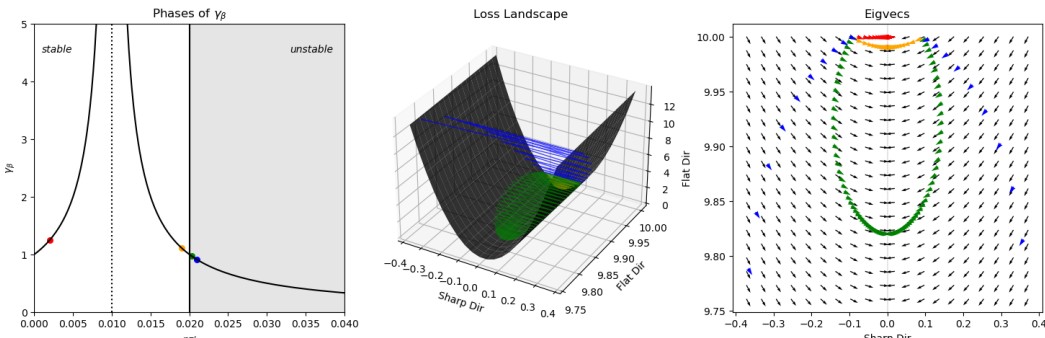

Figure 1: **Optimization trajectories in a $2$-parameter DLN display rotations.** We visualise $4$ our trajectories, initialised at $(-0.1, 10)$ with $z = \Theta^2$ and choosing $\eta \in \{0.001, 0.0095, 0.011, 0.013\}$, where the stability limit (at initialization) is $\eta_{\text{eos}} = 0.01$. **Left:** the regimes of $\gamma_\beta$. **Middle:** the loss landscape. **Right:** map of eigenvector orientations (y-axis magnitudes amplified for clarity).

The regimes of $\gamma_\beta$ can be studied through the behavior of the denominator, $\gamma_\Theta$. The function asymptotes when $\gamma_\Theta \to 0$ at $\eta z' \in \{\theta_1/\theta_2, \theta_2/\theta_1\}$, where the first asymptote matches the sign switch of $\theta_1$ and is exactly half of the stability limit. This guarantees the existence of the first asymptote, and the function reaches $\gamma_\beta = 1$ at:

$$\eta_{\gamma_\beta=1} z' = \frac{2\theta_1\theta_2}{\theta_1^2 + \theta_2^2} = \frac{2\theta_1}{\theta_2}\left(1 - \frac{\theta_1^2}{\theta_1^2 + \theta_2^2}\right)$$
$$= \eta_{\text{eos}}(1 - \epsilon)$$

where we define $\epsilon = \theta_1^2/(\theta_1^2 + \theta_2^2)$. We can then characterize the regimes of $\gamma_\beta$ as a function of $\eta$:

1. From $0 \leq \eta \leq \frac{2\theta_1}{\theta_2 z'}(1 - \epsilon)$ is the **stable** regime of training[1]. This can be further divided into the non-oscillatory and oscillatory regimes by $\frac{2\theta_1}{\theta_2 z'}(1 - \epsilon)$ through the sign of $\theta_1$. In these regime, $\gamma_\beta > 1$, which leads to increases in $R$, signifying increased alignment of the sharpest eigenvector $v_1$ to $\theta_1$, the sharper parameter. As $\eta z' \to \frac{\theta_1}{\theta_2}$, $\gamma_\Theta \to \infty$.

2. As $\eta > \frac{2\theta_1}{\theta_2 z'}(1 - \epsilon)$, we enter the **unstable** regime. In this regime, $\gamma_\beta < 1$, which leads to decreases in $R$, implying that the alignment of the sharpest eigenvector $v_1$ rotates away from $\theta_1$, the sharper parameter.

The existence of the unstable regime along $\theta_1$ requires that the assumption on relative magnitudes is upheld, i.e. $\eta_{\text{eos}} < \eta_{\gamma_\beta=0} = 1/z' \implies \theta_2^2 > 2\theta_1^2$, which is a mild constraint on the relative magnitudes of parameters, especially when considering the ill-conditioning of deep neural networks in practice (Papyan, 2019). In Figure 1, we observe different behaviors in the orientation of the sharpest eigenvector for different choices of $\eta$ (using a quadratic $z$). For stable $\eta$s, the non-oscillatory trajectory ($\eta = 0.001$) takes an almost straight-line path to the local minima at $\theta_1 = 0$ while the oscillatory trajectory ($\eta = 0.0095$) reaches a distant minima with a reduced value in $\theta_2$, though the final eigenvectors are similarly oriented toward $\theta_1$. These minima represent the attractors accessible to different choices of $\eta$s as $\theta_1 \to 0$.

For unstable $\eta$s, the trajectories share a common initial stage during which the sharpest eigenvector moves away from $\theta_1$. While both these trajectories return to stability later on, $\eta = 0.011$ does to as a faster rate. These choices of $\eta$ are able to return to stability in our model because the assumptions imply convexity on the parameters, which means that any optimization step, even during instabilities, leads to optimizations in other parameters which brings us closer toward the global minimum. As the parameter $\Theta$ is adjusted so that loss is improved, $S(\theta)$ translates downwards, which (smoothly) leads to a flat region of parameter space within which stable optimization resumes. These behaviors are mirrored in the general $n$-parameter model through $R_n$ (Eqn. 13).

---

[1] When $z$ is a quadratic function, $\frac{\theta_1}{\theta_2 z'} = 1/\lambda$, so the above thresholds are equal to the $1/\lambda$ and $2/\lambda$ bounds.

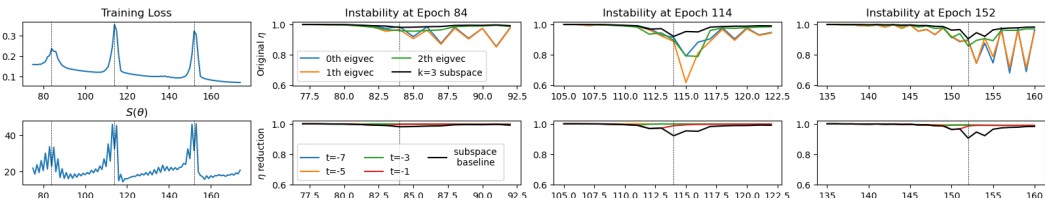

Figure 2: **During instabilities, the sharpest eigenvectors of the Hessian rotate away smoothly and monotonically (Top), while stable training reverts these rotations (Bottom).** We track the similarity of the sharpest Hessian eigenvectors across epochs through three instabilities. **Left:** $L(\theta)$ and $S(\theta)$. **Top:** similarities of the $k$-th eigenvectors (colored) and of subspaces formed by the top 3 eigenvectors (black) during instabilities. **Bottom:** similarity of subspaces formed by the top 3 eigenvectors to the baseline (black) across various timings (colored) when $\eta$ reduction begins.

### 3.2 RESOLUTION OF INSTABILITIES

The DLN model illustrates the gradient descent trajectory for an individual term in the loss function, modelled as a product of individual weights. Crucially, we identified different directions of rotation among the sharpest Hessian eigenvectors, which we build upon to characterize the empirical behavior of deep neural networks during instabilities. We train a multi-layer perceptron (MLP) on fMNIST (Xiao et al., 2017). The rotation of the sharpest Hessian eigenvectors is tracked in Figure 2, and the landscape along the gradient of $L(\theta)$ and $S(\theta)$ is shown in Figure 3.

Figure 2 shows the degree of rotation among the sharpest Hessian eigenvectors at three snapshots of instability, where we compared eigenvector similarities to baselines defined at the beginning of each snapshot. Focusing on the top panel, we observe gradual and monotonic rotations as $L(\theta)$ approaches its peak values as predicted by the DLN model. Notably, these rotations do not involve sudden changes in any single eigenvector but reflect a general decrease in similarity across all eigenvectors, conforming to behaviors of the DLN. Interestingly, even after the instability is resolved, the similarity among individual eigenvectors fall while the subspace comparison remain largely similar. As the subspace of top eigenvectors is heavily constrained by the problem, the observation of re-orientations suggest that new combinations of these eigenvectors can be beneficial toward flatness. In the bottom pane, we intervene to enforce stability by setting $\eta_{\text{low}} = 0.2\eta$s before the instability is resolved. As these models have undergone periods of instability, the eigenvectors are rotated away from their original orientations, but these rotations are immediately reversed once $\eta$ reduction takes place. This reflects an important feature of eigenvectors during instabilities - that stability and instability encourages monotonic rotations in opposite directions.

Although the rotation of eigenvectors follows theoretical expectations, the curvature during the resolution of instabilities displays distinct characteristics. In Figure E, we visualize the landscape of $L(\theta)$ and $S(\theta)$ along the gradient through a complete instability cycle from epoch 119 to 158. As the parameters grow, it moves beyond the region approximated by low-order Taylor Expansions at the minima and the influence of higher-order terms are revealed.

At epoch 119, when $S(\theta) < 2/\eta$, we observe a steady increase in $S(\theta)$ while $L(\theta)$ decreases. This is progressive sharpening. During this phase, the curvature of both $L(\theta)$ and $S(\theta)$ sharpens. While sharpening in $L(\theta)$ is well-documented (as an increase in $S(\theta)$), the sharpening of $S(\theta)$ indicates the increased influence of higher-order terms in the local Taylor-expansion, and that a higher-order derivative is increased (while parameters approach the minima). After epoch 131, when $S(\theta) > 2/\eta$, the curvature of both $L(\theta)$ and $S(\theta)$ remains largely constant, but parameter growth forces the model to explore wider ridges of the local minima along the unstable directions. We expect this to eventually lead to model divergence, but significant changes to the $S(\theta)$ curve are observed after epoch 149, and fortunately a flatter curve is found by epoch 154. At this point, the reduction in $S(\theta)$, as a result of a flatter $S(\theta)$ curve, outweighs the increased steps as a result of large parameters, resulting in optimization steps toward stability in the following epochs while the orientations of the top eigenvectors are reinforced through stability. We observe these effects as a 'cooling' of $L(\theta)$ and $S(\theta)$, before progressive sharpening eventually drives the model back toward instability to repeat the cycle. The full epoch-by-epoch progression is shown in the Appendix E.

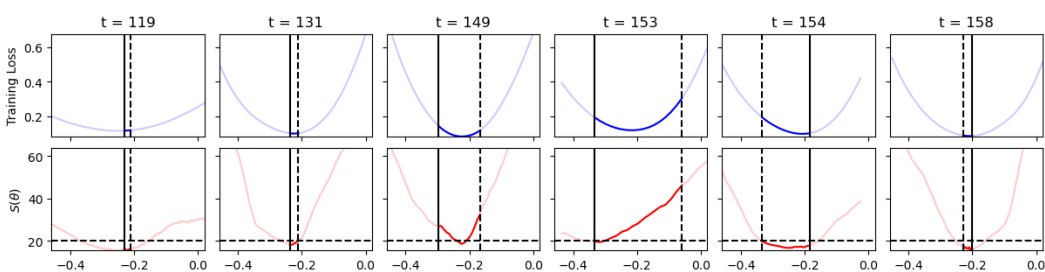

Figure 3: **Parameter growth along the sharpest Hessian eigenvectors leads to exploration of the peripheries of the local minima, driving up $L(\theta)$ and $S(\theta)$ in the process. As the instability develops, the $S(\theta)$ curve undergoes large changes until a flat region is found to enable a return to stability.** We show snapshots along the instability cycle taken along the direction of the gradient. The dotted/solid vertical lines indicate the positions of previous/current parameters, respectively.

Theoretical and empirical evidence suggests that rotations play a crucial role in resolving instabilities. In the DLN model, reductions in $S(\theta)$ were caused by adjustments in $\Theta$ toward the global minimum, which gradually relaxed the stability limit. In contrast, empirical observations showed that $S(\theta)$ decreased quickly and abruptly. Additionally, the post-instability training losses did not significantly differ from trajectories with stable $\eta$s, implying minimal movement of parameters towards minima with lower training loss. However, stable training reinforces the sharpest Hessian eigenvectors while instability disrupts them, creating an implicit bias toward flatness. When $S(\theta) > 2/\eta$, rotations allow the model to explore different orientations of eigenvectors, resulting in varying curvatures of $S(\theta)$. In this process, flat (with respect to $S(\theta)$) eigenvectors are reinforced while sharp ones are discarded. We conjecture that gradient descent, through rotations in Hessian eigenvectors during instability, implicitly biases the parameters toward flatter regions of the loss landscape.

### 3.3 PROGRESSIVE FLATTENING

While a reduction in $S(\theta)$ would be sufficient to achieve stability, our empirical study revealed that instabilities may also reduce higher order derivatives (the curvature of curvatures), leading to a phenomenon we term *progressive flattening*. This effect manifests through a reduction in progressive sharpening, whose effects on the $S(\theta)$ curve are indirect (see Appendix D). The degree of progressive sharpening can be tracked over extended periods of stable training and measuring the eventual maximum sharpness $S(\theta)_{\max}$ that is reached.

In Figure 4, we track $S(\theta)_{\max}$ (for MLPs on fMNIST) as $\eta$ reductions, set to ensure stability, are applied at different stages. $S(\theta)_{\max}$ decreases as $\eta$ reductions are delayed, and larger initial $\eta_0$ used, suggesting that prolonged training with larger learning rates intensifies the regularization effects.

Our findings highlight a strong link between training with large learning rates and the resulting flatness of local parameter space. In Section 4, we use the connection between $S(\theta)$ and generalization to show that large learning rates contribute to improved generalization through *progressive flattening*.

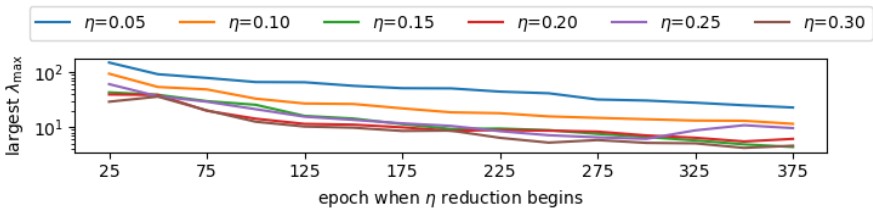

Figure 4: **Progressive flattening in fMNIST.** We plot the eventual maximum $S(\theta)_{\max}$ of MLPs trained with a constant large initial learning rate $\eta_0$ before reducing to $\eta_{\mathrm{small}} = 0.01$ at indicated epochs. The larger and longer phase with $\eta_0$, the more we observe a reduction in $S(\theta)_{\max}$.

## 4 EFFECT ON GENERALIZATION PERFORMANCE

Section 3 highlighted *progressive flattening*, the implicit regularization from instabilities which seeks flatter minima, that can be increased with larger learning rates and with longer duration. In this section, our empirical study validates the benefit of large learning rates toward generalization.

Generalization refers to the ability for neural networks to perform well on data not used in training. The generalization gap is defined as the difference between performance on training data (in-sample) and on unseen test data (out-of-sample). To standardize measurements, models in this section are trained to completion, defined as achieving $> 99\%$ accuracy on the training set. Consequently, test accuracy serves as a direct indicator of the generalization gap.

Some of our experiments in this section are conducted on the CIFAR10 image classification dataset (Krizhevsky, 2009) using small VGG (Simonyan & Zisserman, 2015) networks. While the instability phenomena occur for many choices of error functions (Cohen et al., 2022), we found training to be more stable with cross-entropy loss compared to mean squared error loss, which motivates the use of cross-entropy loss in our experiments. These experiments are performed in a fully non-stochastic setting, with full-batch gradient descent and eschewing common data augmentation techniques such as random flips and crops (Krizhevsky et al., 2012). Additionally, since batch normalization (Ioffe & Szegedy, 2015) benefits deep convolutional architectures, we use the non-stochastic *GhostBatchNorm* (Hoffer et al., 2018) computed over fixed batch size 1024, maintaining the default ordering of data in CIFAR10. To ensure the robustness of our study, we trained hundreds of models, but due to budget constraints our experiments in this section are conducted on a reduced 5k ($10\%$) subset of the full dataset. In Section 4.4, we remove these constraints and benchmark our observations on the full 50k dataset, and introduce non-stochastic augmentations to achieve performance near the state-of-the-art. For further details of our network architecture and experimental setup, see Appendix C.

### 4.1 LARGE LEARNING RATES

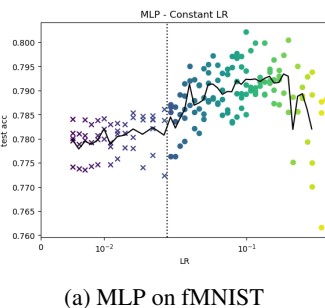 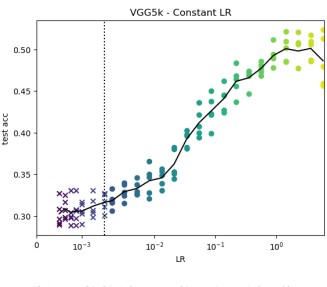

| (a) MLP on fMNIST | (b) VGG10 on CIFAR10-5k |
|---|---|

Figure 5: **Generalization performance improves past the stability limit.** We train models until completion and plot validation accuracy against the learning rate $\eta_0$. The X/O markers differentiate $\eta_0$s below/above the stability limit (dotted line), and the color spectrum from dark purple to light yellow marks the different learning rates from low to high.

We study the effects of learning rates ($\eta_0$s) on generalization performance training MLPs on fMNIST and small VGG10s on CIFAR10-5k. To cover a broad range, $\eta_0$s are sampled on an exponential scale starting with $\eta = [0.01, 0.0005]$ (leading to stable trajectories) and scaling factors $m = [1.1, 1.6]$ for each task, respectively. Sampling continues until models diverge, and each model is tested over 5 random initializations, resulting in a total of $[235, 120]$ models for each task.

Validation accuracy across learning rates are shown in Figure 5. For both tasks, the mean accuracy remains relatively flat until $\eta$ goes past the stability threshold, where it sharply improves. This shift highlights the immediate impact of instabilities, which provide notable generalization benefits, as described in earlier sections. Performance eventually falls, indicating that excessive learning rates can be detrimental. These results suggest a *Goldilocks* zone for learning rates.

The impact on generalization varies between datasets. As CIFAR10 is considered relatively more challenging, regularization plays a more critical role. Nevertheless, the improvements on fMNIST indicate that the regularization effect is present even in simpler, nearly linearly separable problems.

## 4.2 LEARNING RATE REDUCTION

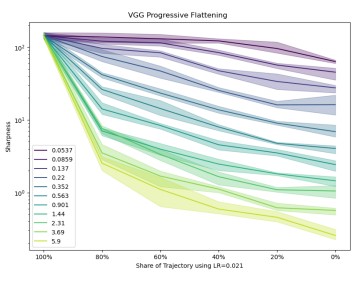 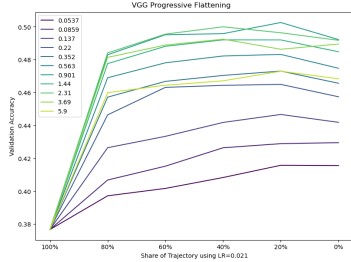

(a) Progressive Flattening  (b) Generalization Performance

Figure 6: **Training with larger learning rates for longer leads to more progressive flattening and improved generalization.** We train small VGG10s on CIFAR10-5k with large initial learning rates, switching to $\eta_{\text{small}} = 0.021$ at various points of training, until completion.

As discussed in Section 3.3, reducing learning rates removes the limit on sharpness to reveal the degree of *progressive flattening*. We begin training small VGG10s with large learning rates $\eta >= 0.086$, which we later reduce to $\eta_{\text{small}} = 0.021$. The final sharpness and validation accuracies are plotted in Figure 6. As large initial learning rates are applied for increasing amounts of time, the at-completion sharpness of models are reduced, which is indicative of *progressive flattening*. For the majority of models, being flatter improves generalization performance. Additionally, the marginal efficiency of large learning rates diminish, but they remain positive for most models.

With *extremely large* learning rates, switching to a lower $\eta$ for the final 20% of training (from $\sim 90\%$ accuracy) may improve generalization. This approach is consistent with the popular choice of learning rate reduction towards the end of training, but the benefits of this strategy can be sensitive to timing.

## 4.3 FLATNESS, ROTATIONS, OR JUST LARGE LEARNING RATES?

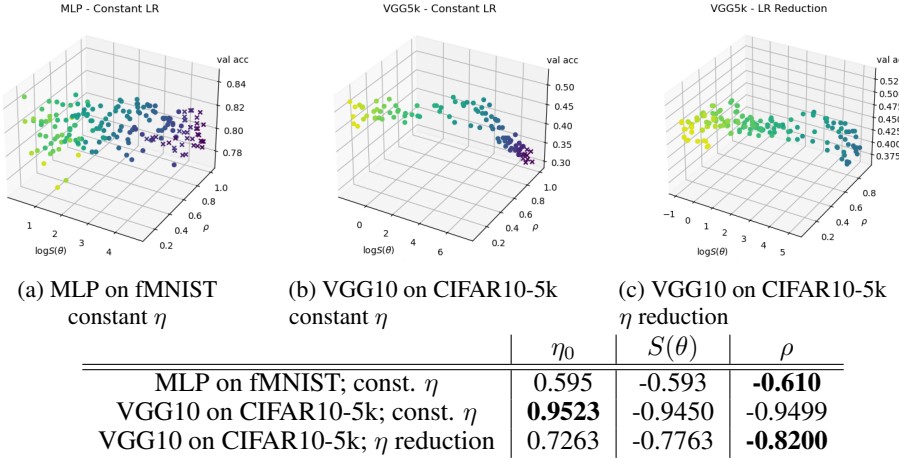

(a) MLP on fMNIST constant $\eta$  (b) VGG10 on CIFAR10-5k constant $\eta$  (c) VGG10 on CIFAR10-5k $\eta$ reduction

| | $\eta_0$ | $S(\theta)$ | $\rho$ |
|---|---|---|---|
| MLP on fMNIST; const. $\eta$ | 0.595 | -0.593 | **-0.610** |
| VGG10 on CIFAR10-5k; const. $\eta$ | **0.9523** | -0.9450 | -0.9499 |
| VGG10 on CIFAR10-5k; $\eta$ reduction | 0.7263 | -0.7763 | **-0.8200** |

Figure 7: **Generalization performance with $S(\theta)$ and $\rho$.** We aggregate models from Figure 5 and 6, adding an additional dimension $\rho$. Rank correlation with validation error are listed in the table below.

Our experiments in Sections 4.1 and Sections 4.2 have demonstrated that the generalization performance of models can be improved through the choice of $\eta_0$s. Larger $\eta_0$s bring about models with lower $S(\theta)$ and more rotations. What are the relative importance of these factors, $\eta_0$, $S(\theta)$, and similarity of eigenvectors $\rho$, toward generalization?

In Figure 7, we outline the association with each metric with validation accuracy, where the evidence demonstrates a clear correlation with generalization for each candidate metric. This effect is more pronounced in CIFAR10 than fMNIST, though across these tasks the strengths of these associations

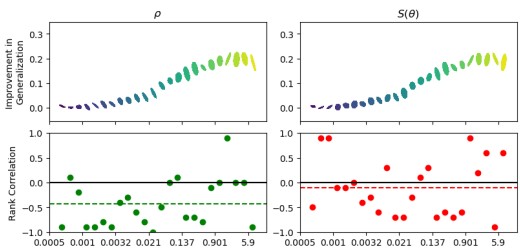

(a) **Left:** $\rho$, **Right:** $S(\theta)$. **Top:** Fit of 2D normal distribution on each set of 5 points, drawn to $1\sigma$. **Bottom:** Rank correlation to generalization, dashed line indicates the mean of correlations.

(b) Mean rank correlation with generalization

|  | corr |
|---|---|
| $\eta_0$ | n.a. |
| $S(\theta)$ | -0.100 |
| $\rho$ | **-0.430** |

Figure 8: **Across weight initialization, $\rho$ is a better predictor than $S(\theta)$ on generalization when $\eta$s is controlled.** We group models of the same $\eta$ into subsets of 5, each differing only by initialization.

are hard to separate under the constant $\eta$ regime, and $\rho$ slightly outperforms alternatives in the $\eta$ reduction regime. With $\eta$ reduction, the models undergo varying degrees of regularization depending on initial learning rates $\eta_0$ and the timing of reduction, with more regularized models tending to prefer stable regimes of training due to progressive flattening. Changes to $S(\theta)$ during stable phases of training are dominated by progressive sharpening, while the orientation of eigenvectors are quickly attracted to fixed points. These differences could explain some of the observed gap.

In our experiments, we explicitly varied learning rates to indirectly control $S(\theta)$ and $\rho$. This approach is equivalent to assuming the causal precedence of $\eta$, which we can leverage to compare the effects of $S(\theta)$ and $\rho$ on generalization. Assuming no additional confounding factors, the isolated effects of $S(\theta)$ and $\rho$ can be measured by controlling for $\eta$. In Figure 8, we group VGG models trained on CIFAR10-5k into sets of 5 that share the same learning rates, where randomness is introduced through weight initialization. The mean correlation of validation accuracy reveals a stronger relationship for $\rho$ than $S(\theta)$, though the overall effect is not conclusive. In this case, the lack of scale-invariance of $S(\theta)$ limits its ability to compare generalization performance across models when learning rates are controlled for. Our findings highlight a scenario where $S(\theta)$ is not effective, while $\rho$ provides a better, albeit inconclusive, predictor of generalization across models.

## 4.4 BENCHMARK ON THE FULL DATASET

Section 4.1 suggested a *Goldilocks* zone for $\eta$ and Section 4.2 suggested that some $\eta$-reduction toward the end of can be beneficial toward generalization. We benchmark these suggestions on the full CIFAR10 dataset, removing constraints imposed on earlier experiments. First, we train small VGG19s on CIFAR10-50k without data augmentations. Next, we introduce non-stochastic data augmentations (crops and flips) by constructing a statically sampled, 10x augmented CIFAR10-500k dataset, which has shown promising results with prior studies (Geiping et al., 2022). Finally, we evaluate ResNet20s (He et al., 2015) on the augmented CIFAR10-500k dataset, with the resulting validation accuracies across various $\eta$s shown in Figure 9.

The evidence suggests improved generalization with large learning rates. For these models, the stability limits for $\eta$ were not computed due to computational constraints, but the chosen values of

| $\eta$ | 0.1 | 0.2 | 0.4 | 0.8 | 1.6 | 3.2 |
|---|---|---|---|---|---|---|
| VGG19 on CIFAR10-50k | 67.09 | 68.36 | 70.53 | 73.41 | 72.70 | 73.56 |
| VGG19 on CIFAR10-500k | 78.12 | 80.63 | 81.73 | 82.88 | 84.01 | 84.07 |
| ResNet20 on CIFAR10-500k | 83.66 | 85.62 | 86.06 | 86.53 | 86.95 | 87.32 |
| ResNet20 on CIFAR10-500k; $\eta_{\text{small}}$ at 90% acc. | n.a. | n.a. | 83.01 | 83.52 | 84.56 | 84.98 |
| ResNet20 on CIFAR10-500k; $\eta_{\text{small}}$ at 98% acc. | n.a. | n.a. | 85.57 | 85.90 | 86.66 | 86.92 |

Figure 9: **Larger learning rates improve generalization on the unconstrained CIFAR10 datasets.** Learning rate reduction with ResNet20s uses $\eta_{\text{small}} = 0.021$ until completion.

$\eta$s were intentionally large to ensure instabilities, as more complex datasets are typically sharper. At $\eta = 6.4$, we observed model divergence, which creates a notable gap between this value and the last functioning learning rate, $\eta = 3.2$, within which the optimal $\eta$ resides. This highlights a limitation of the exponential sampling method used for learning rates. Nevertheless, our results suggest that the *Goldilocks* zone for $\eta$ lies much closer to the divergence boundary than the stability limit - typically within one order of magnitude of the former and several orders of magnitude from the latter. Consequently, in practice, we recommend using learning rates much higher than what is derived from the *descent lemma*.

Finally, we explored learning rate reduction in ResNet20s trained on CIFAR10-500k, switching to $\eta_{\text{low}} = 0.1$ at $90\%$ and $98\%$ training accuracies. The observed decline in performance suggests that finding optimal timing for $\eta$ reductions can be challenging in practice.

## 5 RELATED WORK

Elements of progressive flattening are observed in the literature. Keskar et al. (2017) and Jastrzębski et al. (2018) found that larger learning rates led to more reductions in sharpness which provided benefits toward generalization, while our builds additionally limits these effects to learning rates that induce instabilities. Moreover, we found that varying the duration of instability can result in different degrees of flattening, aligning with Andriushchenko et al. (2023b)'s observation of *loss stabilization*, where phases of large initial learning rates are shown to promote generalization.

Mechanisms toward the resolution of instabilities have also been studied. Lewkowycz et al. (2020) identified a regime of learning rates, above the stability limit, that can *catapult* models into flatter regions. We identify the potential role of rotations in the catapult effect. Damian et al. (2023) showed that, considering one unstable eigenvalue, gradient descent has a tendency to self stabilize due to the cubic term in the local Taylor expansion. While acknowledging the importance of higher-order terms, the effects we identify are present without requiring sharpening factor $\alpha > 0$, which was also rarely observed in empirical studies (see Appendix D). Arora et al. (2022) demonstrated that gradient descent can lead to alignment between gradient the sharpest eigenvector of the Hessian $v_1$. Our model predicts, and we observe, this alignment during stable phases of training when eigenvectors are reinforced, but during instabilities, the orientations of $v_1$ are disrupted.

Our empirical work contributes to the growing body of research toward assessing sharpness as a metric for generalization. While the lack of scale-invariance is a weakness for sharpness Dinh et al. (2017), remedies have been proposed (e.g. (Kwon et al., 2021)). Kaur et al. (2023) suggest that the pathological nature of these transforms may not arise with standard optimizers. However, recent results by Andriushchenko et al. (2023a) on modern benchmarks further challenges the sharpness-generalization link, suggesting that the right measure of sharpness may depend on features of the dataset. Our study indicates that once learning rates are controlled and differentiated through weight initialization, sharpness can fail to discriminate between models. Given the crucial role of Hessian eigenvectors rotations in resolving instabilities, we encourage further exploration of their role in the dynamics of gradient descent.

Lastly, Geiping et al. (2022) demonstrated that state-of-the-art performance on CIFAR10 can be achieved using full-batch gradient descent. Our results, also in a non-stochastic setting, replicate their performance (without explicit regularization), using only learning rates for regularization.

## 6 CONCLUSION

Our work highlights a significant implicit bias in gradient descent that favors flatter minima, an effect frequently conjectured in the deep learning literature. We identify the rotations of Hessian eigenvectors as the primary mechanism driving such exploration, which can lead to *progressive flattening* of the loss landscape. Using learning rates above the stability threshold, deep neural networks can realize substantial benefits in generalization, reinforcing the already prevalent use of large learning rates among practitioners. Our experiments are conducted in a fully non-stochastic setting, and we encourage further work to determine whether these effects can extend to stochastic gradient descent. We hope our work inspires future efforts toward addressing the role of rotations of the Hessian eigenvectors for a better understanding of optimization with gradient descent.

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

## A DETAILS TO THE 2 PARAMETER DLN STUDIED IN SECTION 3

### A.1 FROM THE LOSS HESSIAN TO $R(\beta)$

From Section 3, we have the loss Hessian:

$$H(\Theta) = \begin{bmatrix} \frac{\partial^2 L}{\partial \theta_1^2} & \frac{\partial^2 L}{\partial \theta_1 \theta_2} \\ \frac{\partial^2 L}{\partial \theta_1 \theta_2} & \frac{\partial^2 L}{\partial \theta_2^2} \end{bmatrix} = \begin{bmatrix} z'' \theta_2^2 & z'' \theta_1 \theta_2 + z' \\ z'' \theta_1 \theta_2 + z' & z'' \theta_1^2 \end{bmatrix} \tag{3}$$

We solve the characteristic equation in two dimensions, $Hv = \lambda v$ to get:

$$\lambda_{1,2} = \frac{z''(\theta_1^2 + \theta_2^2) \pm \sqrt{z''^2(\theta_1^2 - \theta_2^2)^2 + 4(z' + z''\Theta)^2}}{2} \tag{4}$$

Assume for now that $\theta_1 > 0$ and $\theta_2 > 0$. We compute the ratio of coordinates:

$$\frac{w_{1,1}}{w_{1,2}} = \frac{\lambda_1 - z''\theta_1^2}{z' + z''\Theta}$$

$$= \frac{z''(\theta_2^2 - \theta_1^2) + \sqrt{z''^2(\theta_2^2 - \theta_1^2)^2 + 4(z' + z''\Theta)^2}}{2(z' + z''\Theta)}$$

From the assumptions on $z$ we get $z'' > 0$. Let us define $r_1 = \theta_2^2 - \theta_1^2$ and $r_2 = \frac{2(z' + z''\Theta)}{z''}$ to get:

$$\frac{w_{1,1}}{w_{1,2}} = g(r_1, r_2) = \frac{r_1 + \sqrt{r_1^2 + r_2^2}}{r_2} \tag{}$$

When $r_1 = 0 < - > \theta_1^2 = \theta_2^2$, we have $w_{1,1} = w1, 2$, which indicates that the sharpest eigenvector is aligned equally to both $\theta$s. The sign of $r_1$ depends on the relative magnitudes of $\theta$s, while the sign of $r_2$ takes the same sign as $\Theta$. We get the conditions:

$$r_1 > 0, 1 < \left| \frac{w_{1,1}}{w_{1,2}} \right| < 1 + \left| \frac{2r_1}{r_2} \right|; r_1 < 0, \left| \frac{w_{1,1}}{w_{1,2}} \right| < 1$$

which shows the sign-indifference of $\left| \frac{w_{1,1}}{w_{1,2}} \right|$ to $r_2$. Since $g(r_1, r_2)g(-r_1, r_2) = 1$, we can get the sign-independent measure of ratios $R$:

$$R(\beta) = \beta + \sqrt{\beta^2 + 1}; \text{ where } \beta = \left| \frac{r_1}{r_2} \right| = \left| \frac{z''(\theta_2^2 - \theta_1^2)}{2(z' + z''\Theta)} \right| \tag{5}$$

### A.2 FROM GRADIENT UPDATES TO $\gamma_\beta$

Let $\Delta x$ denote the update $x_{t+1} = x_t + \Delta x$, and $\gamma_x$ denote the ratio $\frac{x_{t+1}}{x_t} = \frac{x_t + \Delta x}{x_t}$.

$$\Delta \theta_1 = \eta \frac{dz}{d\theta_1} = \eta z' \theta_2; \text{ similarly } \Delta \theta_2 = \eta z' \theta_1$$

$$\gamma_{r_1} = 1 - \eta^2 z'^2$$

$$r_2 = 2(\frac{z'}{z''} + \Theta) = \begin{cases} 2\Theta, & \text{when } x \to 0 \\ c\Theta, & \text{when } x \to \infty, \text{where } 2 < c \leq 4 \text{ is a constant} \end{cases}$$

Given the constant scaling to $\Theta$ in both limits, we approximate the ratio of change $\gamma_{r_2}$ with $\gamma_{r_\Theta}$:

$$\gamma_{r_2} \approx \gamma_\Theta = 1 - k\eta z' + \eta^2 z'^2, \text{ where } k = (\frac{\theta_1}{\theta_2} + \frac{\theta_2}{\theta_1}) > 2$$

$$\implies \gamma_\beta = \frac{\gamma_{r_1}}{\gamma_{r_2}} \approx \frac{\gamma_{r_1}}{\gamma_\Theta} = \frac{1 - \eta^2 z'^2}{1 - \eta z' k + \eta^2 z'^2}$$

## B ROTATIONS IN A GENERAL $n$-PARAMETER DLN

Let loss $L(\Theta)$ be described by $z(\Theta)$, a non-negative convex polynomial with a unique minimum at $\Theta = 0$, limiting $z(\Theta)$ to even-degree polynomials. Further define:

$$\boldsymbol{v}_i := (w_{i,1}, w_{i,2}, ..., w_{i,n})$$

$$D := z''\Theta^2 + z'\Theta$$

$$C_i := \sum_j^n (z''\Theta^2 + z'\Theta)w_{i,j}\theta_j^{-1} = D\sum_j^n w_{i,j}\theta_j^{-1}$$

Using $H\boldsymbol{v}_i = \lambda_i v_i$, we get:

$$H(\Theta)_{jk} = (z''\Theta^2 + z'\Theta)\theta_j^{-1}\theta_k^{-1} - z'\Theta\theta_j^{-1}\theta_k^{-1}\delta(j = k)$$
$$= D\theta_j^{-1}\theta_k^{-1} - z'\Theta\theta_j^{-1}\theta_k^{-1}\delta(j = k)$$

For each coordinate of $\boldsymbol{v}_i$, we have:

$$\lambda_i w_{i,k} = (\sum_j^n Dw_{i,j}\theta_j^{-1}\theta_k^{-1}) - z'\Theta w_{i,k}\theta_k^{-2} \tag{6}$$

$$(\lambda_i + \frac{(z'\Theta)}{\theta_k^2})w_{i,k} = \sum_j^n (z''\Theta^2 + z'\Theta)w_{i,j}\theta_j^{-1}\theta_k^{-1} = \frac{C_i}{\theta_k},$$

$$C_i = (\lambda_i + \frac{z'\Theta}{\theta_k^2})w_{i,k}\theta_k \tag{7}$$

$\lambda_1$ is maximised through $C_1$, which we can maximise with constrained optimization:

$$\text{maximise } F(w_1) = \frac{C_1}{D} = \sum_j^n \frac{w_{1,j}}{\theta_j} \tag{8}$$

$$\text{subject to } \sum_j^n (w_{1,j})^2 - 1 = 0, \quad (\textit{normalization constraint})$$

which is maximised by the solution set of $w_1^*$s:

$$\psi := w_{1,j}^*\theta_j = \frac{1}{\sqrt{\sum_k^n \theta_k^{-2}}} < \theta_1 \tag{9}$$

$$\sum_j^n \frac{w_{1,j}^*}{\theta_j} = \psi\sum_j^n \theta_j^{-}2 = \psi^{-1}$$

The set of $w_1^*$s are fixed for each set of $\theta_j$s, and each pair $w_{1,j}^*\theta_j$ is equal to a constant $\psi$. However, this analytical solution is only an approximation because the constancy constraints were ignored:

$$\text{subject to } \forall j, (\lambda_1 + z'\Theta\theta_j^{-2})w_{1_j}\theta_j = C_1 = \text{const}, \quad (\textit{constancy constraint})$$

Let $\widehat{w}_1^*$s be the solution set with additional constancy constraints. Clearly, $\sum_j^n \frac{\widehat{w}_{1,j}^*}{\theta_j} \leq \sum_j^n \frac{w_{1,j}^*}{\theta_j}$. Let:

$$\widehat{w}_{1,j}^* = w_{1,j}^*(1 + \epsilon_{1,j})$$

Applying the normalization constraint, we get:

$$\sum_j^n (w_{1,j}^{*,2})(1 + \epsilon_{1,j})^2 - 1 = 0 \tag{10}$$

Define $B = \sum^n w_{1,j}^{*,2} \epsilon$ and $C = \sum^n w_{1,j}^{*,2} \epsilon^2$, we substitute in 10 to get:

$$\sum_j^n w_{1,j}^{*,2} + 2B + C - 1 = 0 \rightarrow C = -2B$$

Using *Cauchy-Schwarz*: $B^2 \le (\sum_j^n w_{1,j}^{*,2})C = C \rightarrow -2 \le B \le 0$

$$\sum_j^n \frac{w_{1,j}^*(1 + \epsilon_{1,j})}{\theta_j} = \psi \sum_j^n \frac{1 + \epsilon_{1,j}}{\theta_j^{-2}} = (1 + B)\psi^{-1}$$

We can rewrite the constancy constraints as:

$$\forall j, (\widehat{\lambda}_1 + \frac{z'\Theta}{\theta_j^2})(1 + \epsilon_{1,j})w_{1,j}^*\theta_j = D(1 + B)\psi^{-1}, -2 \le B \le 0 \tag{11}$$

Where we use $\widehat{\lambda}_1$ to denote sharpness that may be different from $\lambda_1$ obtained from $w_{1,j}^*$ (only using the normalization condition). Empirically, we find that $B \rightarrow 0^-$, which is consistent with the maximization setting and that these additional constraints are mild. We now assume that the parameters of the network are ill-conditioned, i.e. $\exists m : \theta_m^2 >> \theta_1^2$, as is commonly observed with deep neural networks (Papyan, 2019) and (Granziol et al., 2021). Using Eqn. 11 we get:

$$j = 0, \ (\widehat{\lambda}_1 + \frac{z'\Theta}{\theta_1^2})(1 + \epsilon_{1,1})w_{1,1}^*\theta_1 = D(1 + B)\psi^{-1}$$

$$j = m : \theta_m^2 >> \theta_1^2, (\widehat{\lambda}_1 + \frac{z'\Theta}{\theta_m^2})(1 + \epsilon_{1,m})w_{1,m}^*\theta_m = D(1 + B)\psi^{-1}$$

$$\rightarrow (\widehat{\lambda}_1 + \frac{z'\Theta}{\theta_1^2})(1 + \epsilon_{1,1})w_{1,1}^*\theta_1 = (\widehat{\lambda}_1 + \frac{z'\Theta}{\theta_m^2})(1 + \epsilon_{1,m})w_{1,m}^*\theta_m \tag{12}$$

$$\rightarrow (\widehat{\lambda}_1 + \frac{z'\Theta}{\theta_0^2}) \approx \widehat{\lambda}_1(1 + \epsilon_{1,m})$$

Using the $j = m$ case, $\widehat{\lambda}_1(1 + \epsilon_{1,m})\psi = D(1 + B)\psi^{-1} \approx D\psi^{-1} = \lambda_1\psi$

$$\rightarrow \widehat{\lambda}_1(1 + \epsilon_{1,m}) = \lambda_1$$

substituting into Eqn 12: $\epsilon_{1,m} \approx \frac{z'\Theta}{\theta_1^2\widehat{\lambda}_1} = \frac{z'\Theta(1 + \epsilon_{1,m})}{\theta_1^2\lambda_1} = \frac{z'\Theta(1 + \epsilon_{1,m})}{\theta_1^2 D\psi^{-2}}$

$$0 < \epsilon_{1,m} < \frac{z'\Theta}{z''\Theta^2}$$

where $0 < \frac{z'\Theta}{z''\Theta^2} \le 1$, taking equality to 1 for a quadratic $z$. Note that this implies:

$$\lambda_1 > \widehat{\lambda}_1 > 0,$$

$$\widehat{\lambda}_1 > \widehat{\lambda}_1\epsilon_{1,m} \approx \frac{z'\Theta}{\theta_1^2}$$

Applying the constancy constraints we can get a ratio of parameters of the sharpest eigenvector, $v_1$:

$$R_n(k) = \left|\frac{w_{1,1}}{w_{1,k>1}}\right| = \frac{\left|\widehat{\lambda}_1\theta_k + z'\Theta/\theta_k\right|}{\left|\widehat{\lambda}_1\theta_1 + z'\Theta/\theta_1\right|}$$

$$= \frac{|\theta_k|\,\widehat{\lambda}_1 + z'\Theta/\theta_k^2}{|\theta_1|\,\widehat{\lambda}_1 + z'\Theta/\theta_1^2} = \frac{f(\theta_k)}{f(\theta_1)} \tag{13}$$

Since $\widehat{\lambda}_1 > 0$ and $z'\Theta > 0$, the function $|\theta|(\widehat{\lambda}_1 + z'\Theta/\theta^2)$ is positive and has positive derivatives when $|\theta| > \theta_{\text{crit}} = (\widehat{\lambda}_1/z'\Theta)^{-0.5}$. The bound $\widehat{\lambda}_1 > z'\Theta/\theta_1^2$ implies $\forall j : |\theta_j| > \theta_{\text{crit}}$, which makes this a monotonically increasing function in $|\theta|$ for our domain, implying $R_n \ge 1$. Additionally, write:

$$\forall k : \theta_k^2 = r_k^2\theta_1^2 \rightarrow \Theta = \theta_1^n \prod_k^n |r_k|, r_k^2 \ge 1$$

By the symmetry of eigenvectors(i.e. multiplying each $\theta$ by a constant should not reorient the Hessian eigenvectors)[2], we have:

$$\frac{dR_n(k)}{d\theta_1} = 0$$

but:

$$\frac{dR_n(k)}{dr_k} = \frac{x f'(r_k\theta_1)f(\theta_1)}{f(\theta_1)^2} > 0$$

The interpretation of $R_n$ is similar to $R(\beta)$ from the $n = 2$ case (Eqn. 5) - a large $R_n$ indicates strong alignment to the unstable parameter, and vice versa. With the above results, we find that $R_n$ yields remarkably similar behavior to $R(\beta)$. Define (relatively) sharp and flat parameters $\theta_s, \theta_f : \theta_f^2/\theta_s^2 = r_{f/s} > 1$, we characterize the phases of learning:

1. In the stable phase of learning, the ratio of parameters $r_{f/s}$ increases, which leads to an increase in $R_n$, This signifies increased alignment of the sharpest eigenvector $\boldsymbol{v}_1$ to $\theta_1$, the unstable parameter.

2. During an instability, the ratio of parameters $r_{f/s}$ falls, leading to a decrease in $R_n$. This implies that the $\boldsymbol{v}_1$ rotates away from $\theta_1$, the unstable parameter

A detailed phase-analysis based on $\eta$ can be conducted with a strategy similar to that employed in the $n = 2$ case.

## C  TECHNICAL DETAILS OF EXPERIMENTS

A large number of experiments were conducted in this work. All of our models are small, considering the large quantity of models trained with an academic budget. These models are trained in a fully non-stochastic setting, using full-batch gradient descent. Moreover, we refrained from the use of common techniques promoting stability, such as momentum and RMSProp (Kingma & Ba, 2017), and explicit regularizers, such as weight decay. We employed the following experimental settings:

1. Figure 1 used a toy 2-parameter DLN model.

2. **MLPs on fMNIST:** The MLPs consisted of 4 hidden layers, each of width 32, for a total of 28480 parameters. This model was trained on $1,000$ samples of fMNIST and evaluated on 200. The dataset was pre-processed with standard normalization. This setting was used in Figures 2, 3, 4, 5a), 7a), 10, 11, and 12.

3. **Small VGG10s on CIFAR10-5k:** The VGG10s consisted of 3 VGG blocks with Ghost-BatchNorm computed at fixed batch size $1,024$ and no dropout. The VGG blocks had 3 convolutional layers, with each block increasing in width $8, 16, 32$, leading to a total of $47,892$ parameters (including BatchNorm params). During training, a fixed 100 epochs were dedicated to a linear warmup schedule for learning rates. This model was trained on $5,000$ samples of CIFAR10 and evaluated on $1,000$, pre-processed with standard normalization. This setting was used in Figures 5 b), 6, 7b) c), 8.

4. **Small VGG19s and ResNet20s on CIFAR10-50k and CIFAR10-500k:** The VGG19s consisted of 6 VGG blocks with 3 convolutional layers, with each block increasing in width $16, 32, 64, 128, 256, 512$, leading to a total parameter count of $335, 277$. The ResNet follows the standard architecture outlined He et al. (2015), where the identity function is used as residual connections as per the original paper, using $271, 117$ parameters. Both models used GhostBatchNorm in place of BatchNorm computed at fixed batch size $1,024$ and no dropout. During training, a fixed 100 epochs were dedicated to a linear warmup schedule for learning rates. These models were trained on $50,000$ and $500,000$ samples of CIFAR10, and evaluated on the full $10,000$ samples available, using the default train-test split and pre-processed with standard normalization. The 500k dataset uses static data augmentations, where each sample from the original 50k is used to generate 10 fixed augmentations, using a combination of random crop and horizontal flips. This setting was used in Figure 9.

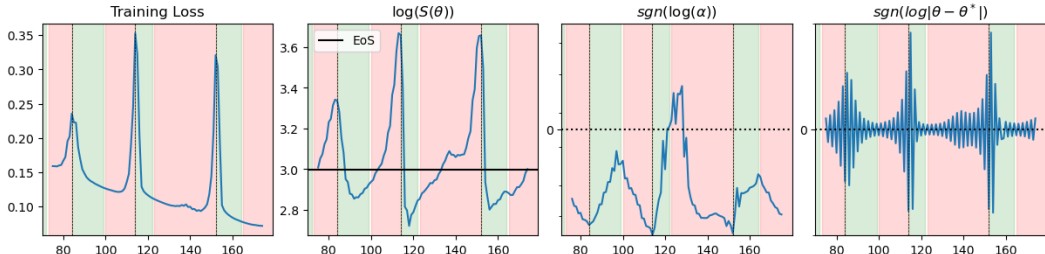

Figure 10: **Magnitude of oscillations divides stable and unstable phases of training.** The turning points of $|\theta - \theta^*|$, distance to mean oscillation parameters, are used to demarcate phases of learning on an MLP trained on fMNIST with FBGD.

## D    DRIVERS OF INSTABILITY

We present an exploration of the training dynamics with gradient descent to pinpoint the sources of instability. The Edge of Stability is reached when $S(\theta)$ rises to $2/\eta$ early in training through progressive sharpening. At this stage, the model exhibits unstable behavior characterized by sudden spikes in training loss and $S(\theta)$, dubbed *instabilities*. To clearly illustrate these dynamics, we train a multi-layer perceptron (MLP) on the fMNIST dataset (Xiao et al., 2017), a simple model on a straightforward task.

Classical theories of stability posits that once past the stability limit, oscillations in parameters become unstable, leading to an increase in magnitude to eventually leads to numerical errors. However, recent findings suggest that deep neural networks can operate at the Edge of Stability for an extended, if not the entire, duration of training. Figure 10 presents a snapshot of a typical training trajectory. Damian et al. (2023) formalised the notion of progressive sharpening by defining a sharpening parameter, $\alpha = -\nabla L(\theta) \cdot \nabla S(\theta)$.[3] This is plotted alongside $|\theta - \theta^*|$, the latter estimating the distance to mean oscillation parameters. In examining the trajectory, we delineate distinct phases of learning through the ascent and descent of $|\theta - \theta^*|$. As gradient descent nears the stability boundary of $2/\eta$, parameter oscillations become more pronounced, driving the system toward instability Notably, the peaks in $L(\theta)$ and $S(\theta)$ coincide with peaks in oscillation magnitude. On the other hand, this contrasts with the progressive sharpening factor, $\alpha$, which remains predominantly negative, indicating that minor gradient adjustments often lead to a reduction in $S(\theta)$, contrary to the expected qualitative effects suggested by progressive sharpening and of instability. This evidence suggests that unstable parameter oscillations, rather than progressive sharpening, are critical drivers of instability.

A more compelling argument arises when we intervene by suppressing parameter updates along unstable directions, setting learning rates to $\eta_u = 0$ specifically for these directions. The modified dynamics, shown in Figure 11, reveal that $S(\theta)$ follows a trajectory similar to that when observed under a globally small learning rate, reflecting a training trajectory completely conducted in the stable regime. This behavior is characterized by slow and gradual increases to $S(\theta)$, which we identify as progressive sharpening, since the latter also exist during stable phases of training. Similarly, reducing the learning rate along unstable directions to achieve *effective stability*, i.e. $\eta_u < 2/S(\theta)$, reproduces this behavior up until the stability threshold is reached, reinforcing the limited role of progressive sharpening toward the formation of instabilities. Conversely, when gradient updates are restricted solely to the directions of the sharpest eigenvectors, unstable oscillations are re-introduced, though the trajectory of $S(\theta)$ significantly changes.

These findings underscore the fundamental differences between progressive sharpening and unstable oscillations. Progressive sharpening primarily describes the increases in $S(\theta)$, the curvature along the sharpest eigenvector, as a consequence of updates in other directions of the Hessian. In contrast, unstable oscillations are driven specifically by parameter updates along these unstable directions. Our results demonstrate that unstable parameter oscillations play a critical role in the formation of instabilities, challenging the view that progressive sharpening plays the primary role. This distinction is important to why the generalization performance of deep neural networks is improved when trained

---

[3]This can be computed with the Hessian trick.

with sufficiently large learning rates (which is observed in Section 4), despite the increases in $S(\theta)$ - for all choices of learning rates - due to progressive sharpening.

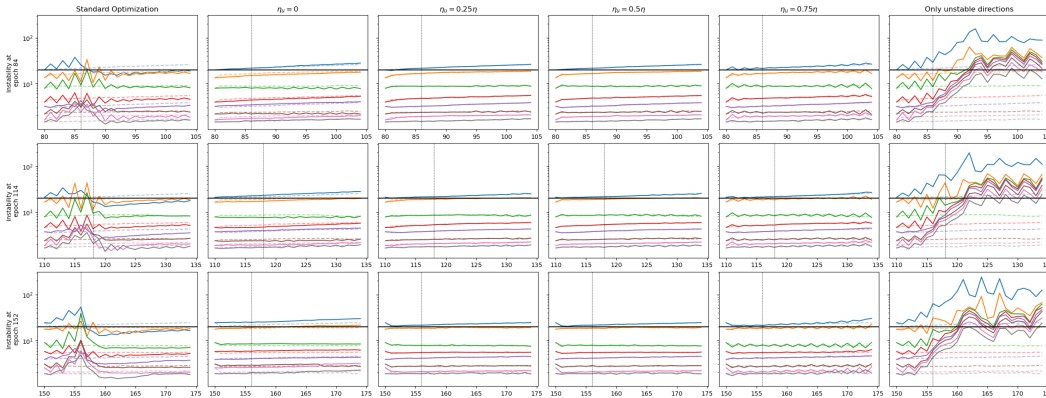

Figure 11: **Limiting step sizes along unstable directions promote stable training, while restricting updates to only unstable directions lead quickly to instability.** We plot the evolution of $S(\theta)$ and sharpness of the top 8 eigenvectors on the same instabilities used in Figure 2. The stable gradient-flow trajectory is plotted with a dashed line.

## E   COMPLETE EVOLUTION OF INSTABILITY VISUALIZED IN FIGURE 3

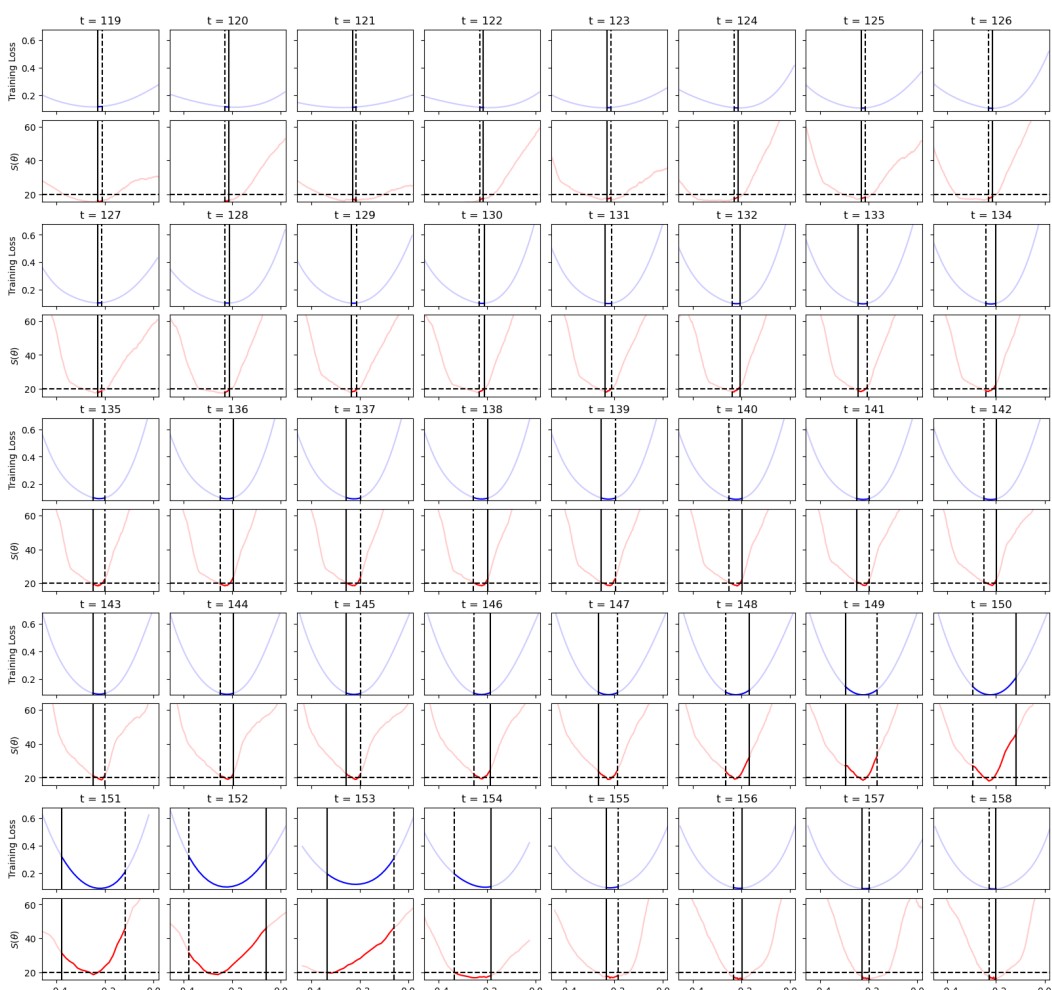

Figure 12: **Parameter growth along the sharpest Hessian eigenvectors leads to exploration of the peripheries of the local minima, driving up $L(\theta)$ and $S(\theta)$ in the process. As the instability develops, the $S(\theta)$ curve undergoes large changes until a flat region is found to enable a return to stability.** We show snapshots along the instability cycle taken along the direction of the gradient. The dotted/solid vertical lines indicate the positions of previous/current parameters, respectively.

