# OpenReview forum: "Can Stability be Detrimental? Better Generalization through Gradient Descent Instabilities"
_ICLR.cc/2025/Conference — ICLR 2025 Conference Withdrawn Submission_

### Official Review · Reviewer_k6KA · 2024-10-31

**Soundness:** 2
**Presentation:** 3
**Contribution:** 3
**Rating:** 5
**Confidence:** 3

**Summary:**

This paper tries to understand the instabilities during neural network training when using large learning rates and their impact on generalization.

The paper theorizes that unstable, high learning rates improve generalization by rotating Hessian eigenvectors, helping the model explore broader regions in the loss landscape. Using a Diagonal Linear Network (DLN) as a toy model, the authors show that unstable parameter growth leads to these beneficial rotations. They confirm this effect experimentally with an MLP on fMNIST, where instabilities cause rotations of the sharpest Hessian eigenvectors.

Then, the authors directly study generalization. They empirically study the impact of large learning rates in different settings (MLP/VGG, CIFAR/MNIST), showing that increasing the learning rate way beyond the theoretical stability limit improves generalization. The authors extend their analysis to decreasing learning rate schedules and try to untangle the impact of different variables correlated with instabilities (learning rate, eigenvector rotation, sharpness), finding that all these variables impact generalization performance, though the eigenvector rotation seems to play a crucial role (compared to sharpness which is often studied in the literature).

**Strengths:**

The paper is well-written. In particular, the theoretical parts (sections 2 and 3) are clear and didactic. Similarly, while I'm not familiar enough with the literature to assess its exhaustivity, I found the related work well-written and easy to follow, giving context to the authors' work.

I did not carefully check the computations but the theoretical parts seem rigorous.

Section 4 provides rigorous experiments. We particularly appreciate that the authors try to empirically understand which specific factor of instabilities impacts generalization the most, and that they provide quite realistic experiments (section 4.4).

**Weaknesses:**

Though it allows for interesting derivations, the DLN model is quite limited, and the paper will be stronger if it studies a more realistic model.

I'm wondering how robust the conclusions in Figure 8 are (see questions).

(Minor) Though I understand that it would require more compute that academic researchers might not have, experiments on harder tasks with more modern models would enrich the paper.

Some figures are quite hard to read. In general, I would recommend increasing the font size, especially for Figures 5 and 6.

For Figure 2: it can seem that the $S(\theta)$ plot is done in the lr reduction setting

For Figure 7, I find the 3d plot hard to read.

For Figure 8, it took me some time to understand what was represented at the top.

For Figure 9, it might be interesting to report the result for lr=6.4 to show that 3.2 is the last stable value.

The code is not currently available, though it seems that it contains interesting contributions (for instance a new Jax implementation of Hessian-Vector product).


### Typos

- Arora et al. (2022) demonstrated that gradient descent can lead to alignment between gradient the sharpest eigenvector of the Hessian
- where exactly derivations are possible

**Questions:**

For Figure 8:
 - Is 5 samples enough to estimate a rank correlation? Could you report the uncertainty in the rank correlation plot?
 - Why does the sharpness often have a high rank correlation with generalization performance, but a varying sign? (compared to rho which has a more consistent but smaller absolute correlation on average).

---

### Official Review · Reviewer_xixY · 2024-10-31

**Soundness:** 2
**Presentation:** 2
**Contribution:** 1
**Rating:** 3
**Confidence:** 4

**Summary:**

This paper studies the "edge of stability" phenomenon, where neural networks can be trained with a larger learning rate than prescribed via the descent lemma. The central claim of this paper is that during instability, the eigenvectors of the Hessian rotate, thus moving the model parameters to a flatter region of the loss landscape -- the authors call this behavior "progressive flattening." The paper provides theoretical support via a diagonal linear network derivation, and shows empirically that the top eigenvectors do rotate during periods of instability, and that large learning rates lead to improved generalization.

**Strengths:**

- It is an important question to understand the dynamics of gradient descent in the large learning rate, or "edge of stability" regime.
- The proposal of eigenvector rotation as a mechanism for reducing instability has not, to the best of my knowledge, appeared before in the literature.
- The theoretical derivations in Section 3 appear to be sound.

**Weaknesses:**

- My main concern with this paper is that it does not adequately support the claim that eigenvector rotation is the *cause* of instabilities being resolved. The paper provides evidence that during instability, the top Hessian eigenvectors rotate, and moreover afterwards the training becomes stable. However, there is no justification that such eigenvector rotation causes the return to stability. Justification for such return to stability has already been given in prior works such as the self-stabilization mechanism of Damian et al., (2023) or the catapult effect of Lewkowycz et al. (2020), and to me it appears that eigenvector rotation is itself caused by instability (yet goes away when other mechanisms cause a return to stability).
- The definition of "progressive flattening" is not clear. My interpretation is that progressive flattening is the phenomenon that higher order derivatives of the sharpness are reduced during instability. However, this effect is not quantified nor easy to observe in the experiments in Section 3.2.
- The justification in Section 3.3 is also insufficient. One alternate hypothesis to Figure 4 that I find more plausible is the following. First sharpness is constrained to $2/\eta$ when training with learning rate $\eta$. Later when $\eta$ is decreased progressive sharpnening occurs, yet the degree of progressive sharpening is proportional to the loss gradient, which is smaller later in training. Therefore reducing $\eta$ later leads to a lower final sharpness.
- Many of the results in Section 4 on the generalization benefit of large learning rates are known in the literature. This benefit is because training with a learning rate of $\eta$ implicitly constrains the sharpness to be at most $2/\eta$; lower sharpness is believed to correlate with better generalization.

**Questions:**

- I find the comparison to Damian et al. (2023) in the Related Work and Appendix D to be inaccurate. That paper proves that, during the unstable regime, 1) the parameters will oscillate and grow in the top eigenvector direction, and 2) via the third-order Taylor expansion, such oscillations will lead to an implicit regularization effect which decreases sharpness and drives the model to stability. This self-stabilization effect occurs whenever training is unstable, and does not require progressive sharpening. The progressive sharpening assumption only characterizes the fact that the sharpness increases during gradient descent for all learning rates, and hence training will eventually becomes unstable. To me, it seems that the "progressive flattening" mechanism presented in this work can be explained by the "self-stabilization" mechanism in Damian et al., (2023).

- Line 244 "Even after the instability is resolved, the similarity among individual eigenvectors fall while the subspace comparison remain largely similar": This doesn't seem to be true in Figure 2, as after the instability the similarity is close to 1.

- Much of the derivation in Section 3.1 is hard to follow. What are the quantities $\gamma_\Theta, \gamma_{|\theta_2^2 - \theta_1^2|}$?Why is the stability limit $\eta_{eos}$ defined to be twice the value of the asymptote?

- In Figure 7, what are these quantities in the table? How is $\rho$ defined? (i.e with respect to what is similarity calculated?)

Minor comments/typos:
- The descent lemma (line 68) does not necessarily require convexity.
- Line 254: "Figure E" -> Figure 12.

---

### Official Review · Reviewer_xkaR · 2024-11-04

**Soundness:** 2
**Presentation:** 1
**Contribution:** 1
**Rating:** 3
**Confidence:** 4

**Summary:**

This paper studies the dynamics of gradient descent in the edge of stability regime. It begins by analyzing gradient descent on a two parameter model $L(\theta) = z(\theta_1 \theta_2)$. It then shows that in realistic settings, the eigenvectors of the Hessian rotate during the EOS dynamics, and conjectures that these rotations are responsible for the edge of stability dynamics. They also propose a "progressive flattening" mechanism which argues that training with large learning rates for longer will lead to flatter solutions, and studies the connection between learning rates, sharpness, eigenvector alignment, and generalization.

**Strengths:**

- The loss landscape visualizations in Figure 3 and Figure 12 are a nice way to visualize the sharpening/flattening process

**Weaknesses:**

- While Figure 2 supports the claim that the eigenvectors of the Hessian rotate at EOS, this is a correlational observation and doesn't support the claim of the paper that this rotation is relevant for stabilizing training, giving rise to the EOS dynamics, or exploring flatter regions of the loss landscape.
- The paper does not precisely define "progressive flattening" – see my question below. In addition, the use of cross entropy acts as a strong confounder in these experiments, as the sharpness will converge to $0$ at the end of training. Therefore, a simple explanation for Figures 4,6 is that using a larger learning rate for longer will decrease the loss more, leading to a smaller overall sharpness. To establish progressive flattening as a distinct phenomenon, it would be valuable to repeat these experiments with MSE loss and train all models to near-zero training loss.
- The paper mischaracterizes the progressive sharpening factor in Damian et al. 2023. Assumption 1 in this paper assumes $\alpha > 0$ along their "constrained trajectory," not the actual gradient descent trajectory. It is expected to be negative along the gradient descent trajectory since the sharpness decreases when self-stabilizing. Their experiments (Appendix E, top left) show $\alpha > 0$ in all of the settings studied in Cohen et al. 2020.
- This paper would benefit from a proper literature review. The only theoretical analyses of EOS cited are Damian et al. 2023 and Arora et al. 2022. There are many relevant missing papers, including papers that study EOS on diagonal linear networks [1,2,3], and [4] which observed a phenomenon possibly related to progressive flattening.

[1] Chen and Bruna 2023: Beyond the edge of stability via two-step gradient updates

[2] Even et al. 2024: (S)GD over Diagonal Linear Networks: Implicit bias, Large Stepsizes and Edge of Stability

[3] Zhu et al. 2023: Understanding Edge-of-Stability Training Dynamics with a Minimalist Example

[4] Kreisler et al. 2023: Gradient Descent Monotonically Decreases the Sharpness of Gradient Flow Solutions in Scalar Networks and Beyond


Minor Points:
- line 154: $theta$s -> $\theta$s

**Questions:**

- The authors claim that instabilities along the top Hessian eigenvector cause the changes in $S$. Is there any evidence for this? It seems that the main changes in sharpness in Figures 3,12 come from the overall curve changing, not from the sharpness differing at different parts of the curve.
- What is the precise definition for progressive flattening?

---

### Official Review · Reviewer_RZmg · 2024-11-06

**Soundness:** 2
**Presentation:** 3
**Contribution:** 2
**Rating:** 5
**Confidence:** 4

**Summary:**

This research paper investigates the relationship between gradient descent instabilities and generalization performance in neural network training. Using toy model and experiments on small scales, they demonstrate that these instabilities cause rotations in the eigenvectors of the loss Hessian.  Through various experiments on benchmark datasets like fMNIST and CIFAR10, the authors provide evidence supporting their claim that larger learning rates can significantly enhance generalization by promoting this implicit regularization effect.

**Strengths:**

* The analysis of Hessian eigenvectors rotation during instabilities is new, best to my knowledge.
* The paper is clearly written and is easy to follow.

**Weaknesses:**

* The authors show that instabilities move training towards flatter regions of the loss landscape. However, This has been shown extensively show in prior works such [1, 4, 5, 6, 7].

* The authors claim that the effect of large learning rates pertains late in training, which they dub it as progressive flattening. First, the phrase can be misunderstood as progressive decrease in sharpness, which has been studied in [4, 5]. I would suggest to use some other phrase for this phenomena. Second, I think the result directly follows from learning rate decay experiments from [9]. If the learning rate is decayed later in training, it would require more time to reach the new EoS threshold. If all models are trained for fixed training time, the sharpness will take more time to increase in the case where learning rate is decayed later in training. I would suggest the authors to train the models long enough such that all models reach the new EoS threshold and then compare the performance.

* The authors claim that their empirical results reveal a clear phase transition where generalization benefits only emerge for
learning rates beyond the stability threshold.  However, looking at Figure 5(b) the transition is not sharp, rather its continuous. I would not recommend using the phrase transition. Also, similar results are already shown in prior works [1, 4, 5, 6].

* The authors claim that they identify that sharpness is not a great predictor of performance. However, this is already known in prior literature, such as [8]. In addition, authors claim that the rotation can be a better predictor than sharpness. But the experimental results are not conclusive.
Moreover, my intuition is that such predictors can be misleading for very flat initializations (Figure 3. of [4]) which do not require instabilities to train at high learning rates. Therefore, I would predict there won't be any significant change in eigenvector rotation.

* The authors claim that they identify that the effects of large learning rate are observed without progressive sharpening. Various prior works have already found that instabilities occur without progressive sharpening [1, 3, 4, 5, 6] and this result is not new. I would like to request the authors to highlight their new claims wrt these works.

* The authors claim that they prove that for any network with depth > 1, instabilities cause rotation along the principle components of Hessian. This is only demonstrated theoretically for deep diagonal networks. This claim should be modified accordingly.

* Similar models to the two parameter model studied in Section 3.1 is extensively analyzed in prior studies, such as [1, 2, 3].
In particular,  when z is quadratic (which is mainly considered in this work) and the global minima is at zero at origin $\theta_1 \theta_2 = 0$ has been analyzed in [1] to demonstrate the catapult effect. As discussed in [3], this model does not exhibit progressive sharpening (also can be seen in equations in [1]), this model does not exhibit progressive sharpening and therefore, this model does not capture real-world behavior well.

[1] The large learning rate phase of deep learning: the catapult mechanism
A Lewkowycz, Y Bahri, E Dyer, J Sohl-Dickstein, G Gur-Ari
arXiv preprint arXiv:2003.02218

[2] Understanding Edge-of-Stability Training Dynamics with a Minimalist Example
Xingyu Zhu, Zixuan Wang, Xiang Wang, Mo Zhou, Rong Ge
ICLR 2023

[3] Universal Sharpness Dynamics in Neural Network Training: Fixed Point Analysis, Edge of Stability, and Route to Chaos
Dayal Singh Kalra, Tianyu He, Maissam Barkeshli
ICLR 2024 BGPT Workshop

[4] Why Warmup the Learning Rate? Underlying Mechanisms and Improvements
DS Kalra, M Barkeshli
arXiv preprint arXiv:2406.09405

[5] Catapults in SGD: spikes in the training loss and their impact on generalization through feature learning
L Zhu, C Liu, A Radhakrishnan, M Belkin
arXiv preprint arXiv:2306.04815

[6] Quadratic models for understanding neural network dynamics
L Zhu, C Liu, A Radhakrishnan, M Belkin
arXiv preprint arXiv:2205.11787

[7] A loss curvature perspective on training instabilities of deep learning models
Gilmer et al.
ICLR 2022

[8] On the maximum hessian eigenvalue and generalization
S Kaur, J Cohen, ZC Lipton
https://arxiv.org/abs/2206.10654

[9] Gradient descent on neural networks typically occurs at the edge of stability
J Cohen, S Kaur, Y Li, JZ Kolter, A Talwalkar
International Conference on Learning Representations

**Questions:**

* In Figure 5, the authors mark the 'stability threshold' using dotted vertical line. Is this stability threshold wrt initialization?
* Can the authors contrast their work with existing literature referred above?

---

### Official Review · Reviewer_yZCU · 2024-11-08

**Soundness:** 3
**Presentation:** 2
**Contribution:** 2
**Rating:** 5
**Confidence:** 2

**Summary:**

Authors study a new phenomenon in training NNs which they call "Progressive flattening" during the instability regime when inverse of learning rate is larger than sharpness. In this regime, they claim, via empirical investigations and a theoretical result about a toy model for diagonal linear network (DLN), that loss goes to flatter regions by changing the eigenvector directions of the Hessian, while in the stable regime it seems the eigenvector directions are reinforced. Experiments show this regularization (starting with large learning rates) and gonig through these instability phases, also has a better effect on generalization They also challenge sharpness as a measure of generalization.
Moreover, their results seem to contradict that of Damian et al 2023 about the behavior of the instability regime.

**Strengths:**

Authors study a new phenomenon about reorientation of eigenvectors of the Hessian during the instability phase of training NNs so that the trajectory can go to flatter regions, which seems new and interesting; even though intuitively it is not clear to me why this is necessary for going to flatter regions. Their claim oppose the claim in Damian et al (2023) that the gradient is well-aligned with the gradient of the sharpness during the instability phase.

**Weaknesses:**

Major concerns:
The relation between “growth of parameters” that the authors discuss with the progressive flattening  and reorienting the eigenvectors, phenomenon is unclear.

The claims seems to oppose the sharpness largest eigenvector alignment with the gradient in the instability regime, so I think the experiments seem insufficient to back their claim, especially about reorienting the eigenvectors

Authors refer to reorienting the eigenvectors of Hessian as the major effect that algorithms goes back to stability, I can imagine reorienting the eigenvectors of the sharpness can have such effect, but the former is not clear to in why it should help. Authors seems not to explain this in their theorem for the toy diagonal model that. Additionally authors interchangeably also talk about the eigenvector of the sharpness Hessian instead of the Hessian of the loss, which is confusing which one is exactly the target of their claims.

Other issues:
033: is dependent —> depends
054:  in generalization —> in the generalization
075: grow without bound —> grow unboundedly
Line 087 definition of $\alpha$ has an issue, there are two dots used after the two gradients
092 last sentence not clear
123: what is a “two-parameter” network?
133: “sharpness hessian eigenvectors” -> you mean loss hessian eigenvectors?
148: very unclear: what does  “sharpest parameter” mean?
157: “despite originating from same loss function” doesn’t make sense, are you referring to different coordinates of gradient being different? Also you are referring to $z$ as loss function, but isn’t that the link function in your notation?

**Questions:**

077 what is the role of this threshold $S(\theta)$ is it on the sharpness? If so we don’t pick it but algorithm trajectory can enforce it, so the sentence “beyond which training is thought to destabilize” is misleading.

Is the theoretical claim about diagonal linear network true for any matrix dimension and depth?

---

### Note · Authors · 2024-11-26

**Comment:**

I thank the reviewers for their helpful feedback. I will work to improve the paper before submitting to a future venue.

**Withdrawal Confirmation:**

I have read and agree with the venue's withdrawal policy on behalf of myself and my co-authors.